# Fast structure learning with modular regularization

**Greg Ver Steeg**
Information Sciences Institute
University of Southern California
Marina del Rey, CA 90292
gregv@isi.edu

**Hrayr Harutyunyan**
Information Sciences Institute
University of Southern California
Marina del Rey, CA 90292
hrayrh@isi.edu

**Daniel Moyer**
Information Sciences Institute
University of Southern California
Marina del Rey, CA 90292
moyerd@usc.edu

**Aram Galstyan**
Information Sciences Institute
University of Southern California
Marina del Rey, CA 90292
galstyan@isi.edu

## Abstract

Estimating graphical model structure from high-dimensional and undersampled data is a fundamental problem in many scientific fields. Existing approaches, such as GLASSO, latent variable GLASSO, and latent tree models, suffer from high computational complexity and may impose unrealistic sparsity priors in some cases. We introduce a novel method that leverages a newly discovered connection between information-theoretic measures and structured latent factor models to derive an optimization objective which encourages *modular* structures where each observed variable has a single latent parent. The proposed method has linear stepwise computational complexity w.r.t. the number of observed variables. Our experiments on synthetic data demonstrate that our approach is the only method that recovers modular structure better as the dimensionality increases. We also use our approach for estimating covariance structure for a number of real-world datasets and show that it consistently outperforms state-of-the-art estimators at a fraction of the computational cost. Finally, we apply the proposed method to high-resolution fMRI data (with more than $10^5$ voxels) and show that it is capable of extracting meaningful patterns.

## 1 Introduction

The ability to recover the true relationships among many variables directly from data is a holy grail in many scientific domains, including neuroscience, computational biology, and finance. Unfortunately, the problem is challenging in high-dimensional and undersampled regimes due to the curse of dimensionality. Existing methods try to address the challenge by making certain assumptions about the structure of the solution. For instance, graphical LASSO, or GLASSO [1], imposes sparsity constraints on the inverse covariance matrix. While GLASSO perfroms well for certain undersampled problems, its computational complexity is cubic in the number of variables, making it impractical for even moderately sized problems. One can improve the scalability by imposing even stronger sparsity constraints, but this approach fails for many real-world datasets that do not have ultra-sparse structure. Other methods such as latent variable graphical LASSO (LVGLASSO) [2] and latent tree modeling methods [3] suffer from high computational complexity as well, whereas approaches like PCA, ICA, or factor analysis have better time complexity but perform very poorly in undersampled regimes.

In this work we introduce a novel latent factor modeling approach for estimating multivariate Gaussian distributions. The proposed method – linear Correlation Explanation or linear CorEx – searches for

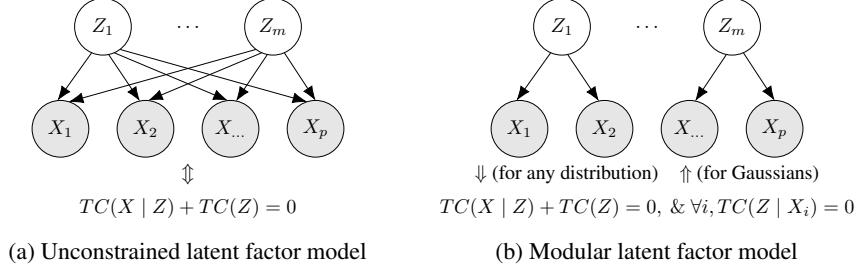

(a) Unconstrained latent factor model          (b) Modular latent factor model

Figure 1: Unconstrained and modular latent factor models. Both models admit equivalent information-theoretic characterization (see Prop. 2.1 and Thm. 2.1 respectively).

independent latent factors that explain all correlations between observed variables, while also biasing the model selection towards modular latent factor models – directed latent factor graphical models where each observed variable has a single latent variable as its only parent. Biasing towards modular latent factor models corresponds to preferring models for which the covariance matrix of observed variables is block-diagonal with each block being a diagonal plus rank-one matrix. This modular inductive prior is appropriate for many real-world datasets, such as stock market, magnetic resonance imaging, and gene expression data, where one expects that variables can be divided into clusters, with each cluster begin governed by a few latent factors and latent factors of different clusters being close to be independent. Additionally, modular latent factors are easy to interpret and are popular for exploratory analysis in social science and biology [4]. Furthermore, we provide evidence that learning the graphical structure of modular latent factor models with fixed number of latent factors gets *easier* as the number of observed variables increases – an effect which we call blessing of dimensionality.

We derive the method by noticing that certain classes of graphical models correspond to global optima of information-theoretic functionals. The information-theoretic optimization objective for learning unconstrained latent factor models is shown in Fig. 1a. We add an extra regularization term that encourages the learned model to have modular latent factors (shown in Fig. 1b). The resulting objective is trained using gradient descent, each iteration of which has *linear* time and memory complexity in the number of observed variables $p$, assuming the number of latent factors is constant.

We conduct experiments on synthetic data and demonstrate that the proposed method is the only one that exhibits a blessing of dimensionality when data comes from a modular (or approximately modular) latent factor model. Based on extensive evaluations on synthetic as well as over fifty real-world datasets, we observe that our approach handily outperforms other methods in covariance estimation, with the largest margins on high dimensional, undersampled datasets. Finally, we demonstrate the scalability of linear CorEx by applying it to high-resolution fMRI data (with more than 100K voxels), and show that the method finds interpretable structures.

## 2  Learning structured models

**Notation**   Let $X \equiv X_{1:p} \equiv (X_1, X_2, \ldots, X_p)$ denote a vector of $p$ observed variables, and let $Z \equiv Z_{1:m} \equiv (Z_1, Z_2, \ldots, Z_m)$ denote a vector of $m$ latent variables. Instances of $X$ and $Z$ are denoted in lowercase, with $x = (x_1, \ldots, x_p)$ and $z = (z_1, \ldots, z_m)$ respectively. Throughout the paper we refer to several information-theoretic concepts, such as differential entropy: $H(X) = -\mathbb{E}\log p(x)$, mutual information: $I(X; Y) = H(X) + H(Y) - H(X, Y)$, multivariate mutual information, historically called total correlation [5]: $TC(X) = \sum_{i=1}^{p} H(X_i) - H(X)$, and their conditional variants, such as $H(X|Z) = \mathbb{E}_z [H(X|Z = z)], TC(X|Z) = \mathbb{E}_z [TC(X|Z = z)]$. Please refer to Cover and Thomas [6] for more information on these quantities.

Consider the latent factor model shown in Fig. 1a, which we call unconstrained latent factor model. In such models, the latent factors explain dependencies present in $X$, since $X_1, \ldots, X_p$ are conditionally independent given $Z$. Thus, learning such graphical models gives us meaningful latent factors. Typically, to learn such a graphical model we would parameterize the space of models with the desired form and then try to maximize the likelihood of the data under the model. An alternative way, the one that we use in this paper, is to notice that some types of directed graphical models can be expressed succinctly in terms of information-theoretic constraints on the joint density function.

In particular, the following proposition provides an information-theoretic characterization of the unconstrained latent factor model shown in Fig. 1a.

**Proposition 2.1.** The random variables $X$ and $Z$ are described by a directed graphical model where the parents of $X$ are in $Z$ and the $Z$'s are independent if and only if $TC(X|Z) + TC(Z) = 0$.

The proof is presented in Sec. A.1. One important consequence is that this information-theoretic characterization gives us a way to select models that are "close" to the unconstrained latent factor model. In fact, let us parametrize $p_W(z|x)$ with a set of parameters $W \in \mathcal{W}$ and get a family of joint distributions $\mathcal{P} = \{p_W(x, z) = p(x)p_W(z|x) : W \in \mathcal{W}\}$. By taking $p_{W^*}(x, z) \in \arg\min_{p_W(x,z) \in \mathcal{P}} TC(Z) + TC(X|Z)$ we select a joint distribution that is as close as possible to satisfy the conditional independence statements corresponding to the unconstrained latent factor model. If for $p_{W^*}(x, z)$ we have $TC(Z) + TC(X|Z) = 0$, then by Prop. 2.1 we have a model where latent variables are independent and explain all dependencies between observed variables. Next, we define modular latent factor models (shown in Fig. 1b) and bias the learning of unconstrained latent factor models towards selecting modular structures.

**Definition 2.1.** A joint distribution $p(x, z)$ with $p$ observed variables $X_{1:p}$ and $m$ hidden variables $Z_{1:m}$ is called modular latent factor model if it factorizes in the following way: $\forall x, z, \; p(x, z) = \left(\prod_{i=1}^p p(x_i|z_{\pi_i})\right)\left(\prod_{j=1}^m p(z_j)\right)$, with $\pi_i \in \{1, 2, \ldots, m\}$.

The motivation behind encouraging modular structures is two-fold. First, modular factor models are easy to interpret by grouping the observed variables according to their latent parent. Second, modular structures are good candidates for beating the curse of dimensionality. Imagine increasing the number of observed variables while keeping the number of latent factors fixed. Intuitively, we bring more information about latent variables, which should help us to recover the structure better. We get another hint on this when we apply a technique from Wang et al. [7] to lower bound the sample complexity of recovering the structure of a Gaussian modular latent factor model. We establish that the lower bound decreases as we increase $p$ keeping $m$ fixed (refer to Sec. C for more details). For more general models such as Markov random fields, the sample complexity grows like $\log p$ [7].

To give an equivalent information-theoretic characterization of modular latent factor models hereafter we focus our analysis on multivariate Gaussian distributions.

**Theorem 2.1.** A multivariate Gaussian distribution $p(x, z)$ is a modular latent factor model if and only if $TC(X|Z) + TC(Z) = 0$ and $\forall i, TC(Z|X_i) = 0$.

The proof is presented in Sec. A.2. Besides characterizing modular latent factor models, this theorem gives us an information-theoretic criterion for selecting more modular joint distributions. The next section describes the proposed method which uses this theorem to bias the model selection procedure towards modular solutions.

## 3 Linear CorEx

We sketch the main steps of the derivation here while providing the complete derivation in Sec. B. The first step is to define the family of joint distributions we are searching over by parametrizing $p_W(z|x)$. If $X_{1:p}$ is Gaussian, then we can ensure $X_{1:p}, Z_{1:m}$ are jointly Gaussian by parametrizing $p_W(z_j|x) = \mathcal{N}(w_j^T x, \eta_j^2), \; w_j \in \mathbb{R}^p, j = 1..m$, or equivalently by $z = Wx + \epsilon$ with $W \in \mathbb{R}^{m \times p}, \epsilon \sim \mathcal{N}(0, \text{diag}(\eta_1^2, \ldots, \eta_m^2))$. W.l.o.g. we assume the data is standardized so that $\mathbb{E}[X_i] = 0, \mathbb{E}[X_i^2] = 1$. Motivated by Thm. 2.1, we will start with the following optimization problem:

$$\underset{W}{\text{minimize}} \; TC(X|Z) + TC(Z) + \sum_{i=1}^p Q_i, \tag{1}$$

where $Q_i$ are regularization terms for encouraging modular solutions (i.e. encouraging solutions with smaller value of $TC(Z|X_i)$). We will later specify this regularizer as a non-negative quantity that goes to zero in the case of exactly modular latent factor models. After some calculations for Gaussian random variables and neglecting some constants, the objective simplifies as follows:

$$\underset{W}{\text{minimize}} \sum_{i=1}^p \left(1/2 \log \mathbb{E}\left[(X_i - \mu_{X_i|Z})^2\right] + Q_i\right) + \sum_{j=1}^m 1/2 \log \mathbb{E}\left[Z_j^2\right], \tag{2}$$

**Algorithm 1** Linear CorEx. Implementation is available at `https://github.com/hrayrhar/T-CorEx`

---

**Input:** Data matrix $X \in \mathbb{R}^{n \times p}$, with $n$ iid samples of vectors in $R^p$.
**Result:** Weight matrix, $W$, optimizing (3).
Subtract mean and scale from each column of data
Initialize $W_{j,i} \sim \mathcal{N}(0, 1/\sqrt{p})$
**for** $\epsilon$ in $[0.6, 0.6 \cdot 0.6^3, 0.6^4, 0.6^5, 0.6^6, 0]$ **do**
    **repeat**
        $\bar{X} = \sqrt{1 - \epsilon^2}X + \epsilon E$, with $E \in \mathbb{R}^{n \times p}$ and $E_{i,j} \overset{\text{iid}}{\sim} \mathcal{N}(0,1)$
        Let $\hat{J}(W)$ be the empirical version of (3) with $X$ replaced by $\bar{X}$
        Do one step of ADAM optimizer to update $W$ using $\nabla_W \hat{J}(W)$
    **until** until convergence or maximum number of iterations is reached
**end for**

---

where $\mu_{X_i|Z} = \mathbb{E}_{X_i|Z}[X_i|Z]$. For Gaussians, calculating $\mu_{X_i|Z}$ requires a computationally undesirable matrix inversion. Instead, we will select $Q_i$ to eliminate this term while also encouraging modular structure. According to Thm. 2.1, modular models obey $TC(Z|X_i) = 0$, which implies that $p(x_i|z) = p(x_i)/p(z) \prod_j p(z_j|x_i)$. Let $\nu_{X_i|Z}$ be the conditional mean of $X_i$ given $Z$ under such factorization. Then we have

$$\nu_{X_i|Z} = \frac{1}{1+r_i} \sum_{j=1}^{m} \frac{Z_j B_{j,i}}{\sqrt{\mathbb{E}\left[Z_j^2\right]}}, \text{with } R_{j,i} = \frac{\mathbb{E}\left[X_i Z_j\right]}{\sqrt{\mathbb{E}\left[X_i^2\right]\mathbb{E}\left[Z_j^2\right]}}, B_{j,i} = \frac{R_{j,i}}{1-R_{j,i}^2}, r_i = \sum_{j=1}^{m} R_{j,i} B_{j,i}.$$

If we let

$$Q_i = \frac{1}{2} \log \frac{\mathbb{E}\left[(X_i - \nu_{X_i|Z})^2\right]}{\mathbb{E}\left[(X_i - \mu_{X_i|Z})^2\right]} = \frac{1}{2} \log \left( 1 + \frac{\mathbb{E}\left[(\mu_{X_i|Z} - \nu_{X_i|Z})^2\right]}{\mathbb{E}\left[(X_i - \mu_{X_i|Z})^2\right]} \right) \geq 0,$$

then we can see that this regularizer is always non-negative and is zero exactly for modular latent factor models (when $\mu_{X_i|Z} = \nu_{X_i|Z}$). The final objective simplifies to the following:

$$\underset{W}{\text{minimize}} \sum_{i=1}^{p} {}^1\!/_2 \log \mathbb{E}\left[(X_i - \nu_{X_i|Z})^2\right] + \sum_{j=1}^{m} {}^1\!/_2 \log \mathbb{E}\left[Z_j^2\right]. \tag{3}$$

This objective depends on pairwise statistics and requires no matrix inversion. The global minimum is achieved for modular latent factor models. The next step is to approximate the expectations in the objective (3) with empirical means and optimize it with respect to the parameters $W$. After training the method we can interpret $\hat{\pi}_i \in \arg\max_j I(Z_j; X_i)$ as the parent of variable $X_i$. Additionally, we can estimate the covariance matrix of $X$ the following way:

$$\widehat{\Sigma}_{i,\ell \neq i} = \frac{(B^T B)_{i,\ell}}{(1+r_i)(1+r_\ell)}, \quad \widehat{\Sigma}_{i,i} = 1. \tag{4}$$

We implement the optimization problem (3) in PyTorch and optimize it using the ADAM optimizer [8]. In empirical evaluations, we were surprised to see that this update worked better for identifying weak correlations in noisy data than for very strong correlations with little or no noise. We conjecture that noiseless latent factor models exhibit stronger curvature in the optimization space leading to sharp, spurious local minima. We implemented an annealing procedure to improve results for nearly deterministic factor models. The annealing procedure consists of rounds, where at each round we pick a noise amount, $\epsilon \in [0, 1]$, and in each iteration of that round replace $X$ with its noisy version, $\bar{X}$, computed as follows: $\bar{X} = \sqrt{1 - \epsilon^2}X + \epsilon E$, with $E \sim \mathcal{N}(0, I_p)$. It can be easily seen that when $\mathbb{E}[X_i] = 0$, and $\mathbb{E}\left[X_i^2\right] = 1$, we get that $\mathbb{E}\left[\bar{X}_i\right] = 0$, $\mathbb{E}\left[\bar{X}_i^2\right] = 1$, and $\mathbb{E}\left[\bar{X}_i \bar{X}_j\right] = (1 - \epsilon^2)\mathbb{E}\left[X_i X_j\right] + \epsilon^2 \delta_{i,j}$. This way adding noise weakens the correlations between observed variables. We train the objective (3) for the current round, then reduce $\epsilon$ and proceed into the next round retaining current values of parameters. We do 7 rounds with the following schedule for $\epsilon$, $[0.6^1, 0.6^2, \ldots, 0.6^6, 0]$. The final algorithm is shown in Alg. 1. Our implementation is available at `https://github.com/hrayrhar/T-CorEx`.

The only hyperparameter of the proposed method that needs significant tuning is the number of hidden variables, $m$. While one can select it using standard validation procedures, we observed that

it is also possible to select it by increasing $m$ until the gain in modeling performance, measured by log-likelihood, is insignificant. This is due to the fact that setting $m$ to a larger value than needed has no effect on the solution of problem (3) as the method can learn to ignore the extra latent factors.

The stepwise computational complexity of linear CorEx is dominated by matrix multiplications of an $m \times p$ weight matrix and a $p \times n$ data matrix, giving a computational complexity of $O(mnp)$. This is only linear in the number of observed variables assuming $m$ is constant, making it an attractive alternative to standard methods, like GLASSO, that have at least cubic complexity. Furthermore, one can use GPUs to speed up the training up to 10 times. The memory complexity of linear CorEx is $O((mT + n)p)$. Fig. 4 compares the scalability of the proposed method against other methods.

## 4 Experiments

In this section we compare the proposed method against other methods on two tasks: learning the structure of a modular factor model (i.e. clustering observed variables) and estimation of covariance matrix of observed variables. Additionally, we demonstrate that linear CorEx scales to high-dimensional datasets and finds meaningful patterns. We present the essential details on experiments, baselines, and hyperparameters in the main text. The complete details are presented in the appendix (see Sec. D).

### 4.1 Evidence of blessing of dimensionalty

We start by testing whether modular latent factor models allow better structure recovery as we increase dimensionality. We generate $n = 300$ samples from a modular latent factor model with $p$ observed variables, $m = 64$ latent variables each having $p/m$ children, and additive white Gaussian noise channel from parent to child with fixed signal-to-noise ratio $s = 0.1$. By setting $s = 0.1$ we focus our experiment in the regime where each individual variable has low signal-to-noise ratio. Therefore, one should expect poor recovery of the structure when $p$ is small. In fact, the sample complexity lower bound of Thm. C.1 tells us that in this setting any method needs at least 576 observed variables for recovering the structure with $\epsilon = 0.01$ error probability. As we increase $p$, we add more weakly correlated variables and the overall information that $X$ contains about $Z$ increases. One can expect that some methods will be able to leverage this additional information.

As recovering the structure corresponds to correctly clustering the observed variables, we consider various clustering approaches. For decomposition approaches like factor analysis (FA) [9], non-negative matrix factorization (NMF), probabilistic principal component analysis (PCA) [10], sparse PCA [11, 12] and independent component analysis (ICA), we cluster variables according to the latent factor whose weight has the maximum magnitude. As factor analysis suffers from an unidentifiability problem, we do varimax rotation (FA+V) [13] to find more meaningful clusters. Other clustering methods include k-means, hierarchical agglomerative clustering using Euclidean distance and the Ward linkage rule (Hier.), and spectral clustering (Spec.) [14]. Finally, we consider the latent tree modeling (LTM) method [15]. Since information distances are estimated from data, we use the "Relaxed RG" method. We slightly modify the algorithm to use the same prior information as other methods in the comparison, namely, that there are exactly $m$ groups and observed nodes can be siblings, but not parent and child. We measure the quality of clusters using the adjusted Rand index (ARI), which is adjusted for chance to give 0 for a random clustering and 1 for a perfect clustering. The left part of Fig. 2 shows the clustering results for varying values of $p$. While a few methods marginally improve as $p$ increases, only the proposed method approaches perfect reconstruction.

We find that this blessing of dimensionality effect persists even when we violate the assumptions of a modular latent factor model by correlating the latent factors or adding extra parents for observed variables. For correlating the latent factors we convolve each $Z_i$ with two other random latent factors. For adding extra parents, we randomly sample $p$ extra edges from a latent factor to a non-child observed variable. By this we create on average one extra edge per each observed variable. In both modifications to keep the the notion of clusters well-defined, we make sure that each observed variable has higher mutual information with its main parent compared to other factors. All details about synthetic data generation are presented in Sec. E. The right part of the Fig. 2 demonstrates that the proposed method improves the results as $p$ increases even if the data is not from a modular latent factor model. This proves that our regularization term for encouraging modular structures is indeed effective and leads to such structures (more evidence on this statement are presented in Sec. F.1).

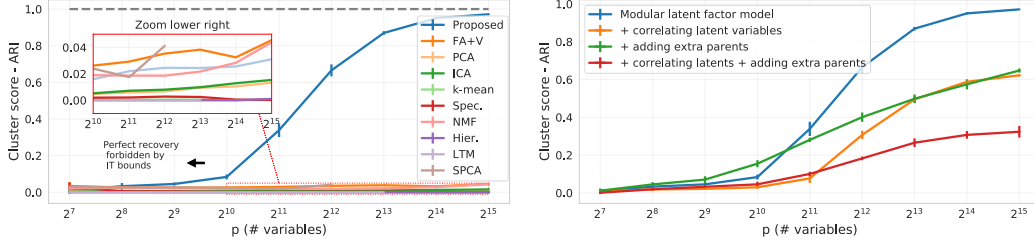

Figure 2: Evidence of blessing of dimensionality effect when learning modular (on the left) or approximately modular (on the right) latent factor models. We report adjusted Rand index (ARI) measured on $10^4$ test samples. Error bars are standard deviation over 20 runs.

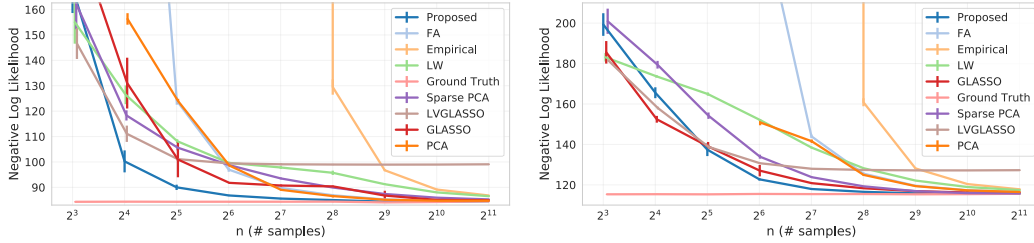

Figure 3: Comparison of covariance estimation baselines on synthetic data coming from modular latent models. On the left: $m = 8$ latent factors each having 16 children, on the right: $m = 32$ latent factors each having 4 children. The reported score is the negative log-likelihood (lower better) on a test data with 1000 samples. Error bars are standard deviation over 5 runs. We jitter $x$-coordinates to avoid overlaps.

## 4.2 Covariance estimation

We now investigate the usefulness of our proposed approach for estimating covariance matrices in the challenging undersampled regime where $n \ll p$. For comparison, we include the following baselines: the empirical covariance matrix, Ledoit-Wolf (LW) method [16], factor analysis (FA), sparse PCA, graphical lasso (GLASSO), and latent variable graphical lasso (LVGLASSO). To measure the quality of covariance matrix estimates, we evaluate the Gaussian negative log-likelihood on a test data. While the Gaussian likelihood is not the best evaluation metric for non-Gaussian data, we would like to note that our comparisons of baselines are still fair, as most of the baselines, such as [latent variable] GLASSO, [sparse] PCA, are derived under Gaussian assumption. In all experiments hyper-parameters are selected from a grid of values using a 3-fold cross-validation procedure.

**Synthetic data** We first evaluate covariance estimation on synthetic data sampled from a modular latent factor model. For this type of data, the ground truth covariance matrix is block-diagonal with each block being a diagonal plus rank-one matrix. We consider two cases: 8 large groups with 16 variables in each block and 32 small groups with 4 variables in each block. In both cases we set the signal-to-noise ratio $s = 5$ and vary the number of samples. The results for both cases are shown in Fig. 3. As expected, the empirical covariance estimate fails when $n \leq p$. PCA and factor analysis are not competitive in cases when $n$ is small, while LW nicely handles those cases. Methods with sparsity assumptions: sparse PCA, GLASSO, LVGLASSO, do well especially for the second case, where the ground truth covariance matrix is very sparse. In most cases the proposed method performs best, only losing narrowly when $n \leq 16$ samples and the covariance matrix is very sparse.

**Stock market data** In finance, the covariance matrix plays a central role for estimating risk and this has motivated many developments in covariance estimation. Because the stock market is highly non-stationary, it is desirable to estimate covariance using only a small number of samples consisting of the most recent data. We considered the weekly percentage returns for U.S. stocks from January 2000 to January 2017 freely available on `http://quandl.com`. After excluding stocks that did not have returns over the entire period, we were left with 1491 companies. We trained on $n$ weeks of data to learn a covariance matrix using various methods then evaluated the negative log-likelihood on the subsequent 26 weeks of test data. Each point in Fig. 5 is an average from rolling

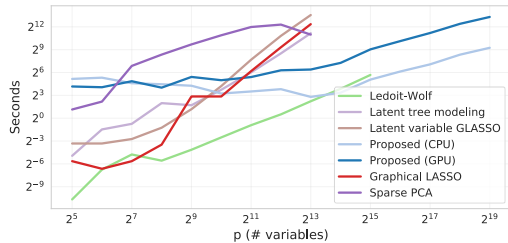

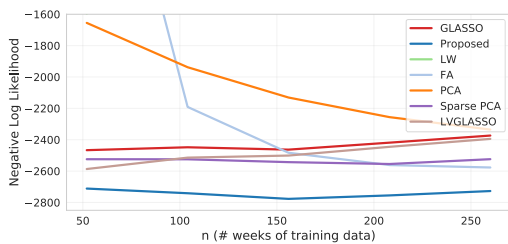

Figure 4: Runtime comparison of various methods. Points that do not appear either timed out at $10^4$ seconds or ran out of memory. The experiment was done in the setting of Sec. 4.1 on an Intel Core i5 processor with 4 cores at 4Ghz and 64Gb memory. We used Nvidia RTX 2080 GPU when running the proposed method on a GPU.

Figure 5: Comparison of covariance estimation baselines on stock market data. The reported score is the negative log-likelihood (lower better) on a test data. Most of the Ledoit-Wolf points are above the top of the $y$ axis.

Table 1: For the first ten latent factors, we give the top three stocks ranked by mutual information between stock and associated latent factor.

| Factor | Stock ticker | Sector/Industry |
|---|---|---|
| 0 | RF, KEY, FHN | Bank holding (NYSE, large cap) |
| 1 | ETN, IEX, ITW | Industrial machinery |
| 2 | GABC, LBAI, FBNC | Bank holding (NASDAQ, small cap) |
| 3 | SPN, MRO, CRZO | Oil & gas |
| 4 | AKR, BXP, HIW | Real estate investment trusts |
| 5 | CMS, ES, XEL | Electric utilities |
| 6 | POWI, LLTC, TXN | Semiconductors |
| 7 | REGN, BMRN, CELG | Biotech pharmaceuticals |
| 8 | BKE, JWN, M | Retail, apparel |
| 9 | DHI, LEN, MTH | Homebuilders |

the training and testing sets over the entire time period. For component-based methods (probabilistic PCA, sparse PCA, FA, proposed method) we used 30 components. We omitted empirical covariance estimation since all cases have $n < p$. We see that Ledoit-Wolf does not help much in this regime. With enough samples, PCA and FA are able to produce competitive estimates. Methods with sparsity assumptions, such as GLASSO, LVGLASSO, and sparse PCA, perform better. We see that LVGLASSO consistently outperforms GLASSO, indicating that stock market data is better modeled with latent factors. The proposed method consistently outperforms all the other methods. Our approach leverages the high-dimensional data more efficiently than standard factor analysis. The stock market is not well modeled by sparsity, but attributing correlations to a small number of latent factors appears to be effective.

To examine the interpretability of learned latent factors, we used weekly returns from January 2014 to January 2017 for training. This means we used only 156 samples and 1491 variables (stocks). For each factor, we use the mutual information between a latent factor and stock to rank the top stocks related to a factor. We summarize the top stocks for other latent factors in Table 1. Factor 0 appears to be not just banking related, but more specifically bank holding companies. Factor 5 has remarkably homogeneous correlations and consists of energy companies. Factor 9 is specific to home construction.

**OpenML datasets**    To demonstrate the generality of our approach, we show results of covariance estimation on 51 real-world datasets. To avoid cherry-picking, we selected datasets from OpenML [17] according to the following criteria: between 100 and 11000 numeric features, at least twenty samples but fewer samples than features (samples with missing data were excluded), and the data is not in a sparse format. These datasets span many domains including gene expression, drug design, and mass spectrometry. For factor-based methods including our own, we chose the number of factors from the set $m \in \{5, 20, 50, 100\}$ using 3-fold cross-validation. We use an 80-20 train-test split, learning a covariance matrix from training data and then reporting the negative log-likelihood on test data. We standardized the data columns to have zero mean and unit variance. Numerical

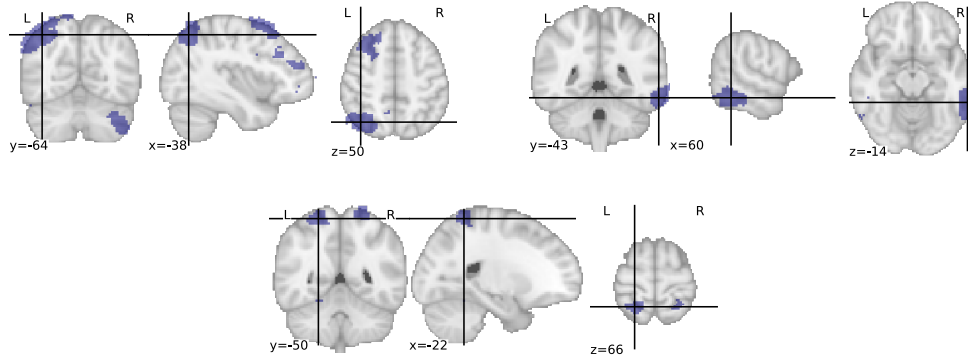

Figure 6: Some of the clusters linear CorEx finds. The cross-hairs correspond to the specified regions.

problems involving infinite log-likelihoods can arise in datasets which are low rank because of duplicate columns, for example. We add Gaussian noise with variance $10^{-12}$ to avoid this.

We compared the same methods as before with three changes. We omitted empirical covariance estimation since all cases have $n < p$. We also omitted LVGLASSO as it was too slow on datasets having about $10^4$ variables. The standard GLASSO algorithm was also far too slow for these datasets. Therefore, we used a faster version called BigQUIC [18]. For GLASSO, we considered sparsity hyper-parameters $\lambda \in \{2^0, 2^1, 2^2, 2^3\}$. We intended to use a larger range of sparsity parameters but the speed of BigQUIC is highly sensitive to this parameter. In a test example with $10^4$ variables, the running time was 130 times longer if we use $\lambda = 0.5$ versus $\lambda = 1$. Due to space limits we present the complete results in the appendix (Sec. F.2, Table 2). The proposed method clearly outperformed the other methods, getting the best score on 32 out of 51 datasets. Ledoit-Wolf also performed well, getting the best results on 18 out of 51 datasets. Even when the proposed method was not the best, it was generally quite close to the best score. The fact that we had to use relatively large sparsity parameters to get reasonable running time may have contributed to BigQUIC's poor performance.

### 4.3 High-resolution fMRI data

The low time and memory complexity of the proposed method allows us to apply it on extremely high-dimensional datasets, such as functional magnetic resonance images (fMRI), common in human brain mapping. The most common measurement in fMRI is Blood Oxygen Level-Dependent (BOLD) contrast, which measures blood flow changes in biological tissues ("activation"). In a typical fMRI session hundreds of high-resolution brain images are captured, each having 100K-600K volumetric pixels (voxels). We demonstrate the scalability and interpretability of linear CorEx by applying it with 100 latent factors on the resting-state fMRI of the first session (session id: 014) of the publicly available MyConnectome project [19]. The session has 518 images each having 148262 voxels. We do spatial smoothing by applying a Gaussian filter with fwhm=8mm, helping our model to pick up the spatial information faster. Without spatial smoothing the training is unstable and we suspect that more samples are needed to train the model. We assign each voxel to the latent factor that has the largest mutual information with it, forming groups by each factor.

Fig. 6 shows three clusters linear CorEx finds. Though appearing fragmented, the cluster on the left actually captures exactly a memory and reasoning network from cognitive science literature [20]. This includes the activations in the Left Superior Parietal Lobule, the Left Frontal Middle and Superior Gyri, and the Right Cerebellum. Though the authors of [20] are describing activations during a task-based experiment, the correlation of these regions during resting state is unsurprising if they indeed have underlying functional correlations. The cluster in the middle is, with a few outlier exceptions, a contiguous block in the Right Medial Temporal cortex. This demonstrates the extraction of lateralized regions. The cluster on the right is a bilateral group in the Superior Parietal Lobules. Bilateral function and processing is common for many cortical regions, and this demonstrates the extraction of one such cluster.

## 5 Related work

Pure one factor models induce relationships among observed variables that can be used to detect latent factors [21, 22]. Tests using relationships among observed variables to detect latent factors have been adapted to the modeling of latent trees [15, 23]. Besides tree-like approaches or pure one factor models, another line of work imposes sparsity on the connections between latent factors and observed variables [11, 12]. Another class of latent factor models can be cast as convex optimization problems [24, 25]. Unfortunately, the high computational complexity of these methods make them completely infeasible for the high-dimensional problems considered in this work.

While sparse methods and tractable approximations have enjoyed a great deal of attention [1, 26–28, 18, 29, 30], marginalizing over a latent factor model does not necessarily lead to a sparse model over the observed variables. Many highly correlated systems, like financial markets [31], seem better modeled through a small number of latent factors. The benefit of adding more variables for learning latent factor models is also discussed in [32].

Learning through optimization of information-theoretic objectives has a long history focusing on mutual information [33–35]. Minimizing $TC(Z)$ is well known as ICA [36, 37]. The problem of minimizing $TC(X|Z)$ is less known but related to the Wyner common information [38] and has also been recently investigated as an optimization problem [39]. A similar objective was used in [40] to model *discrete* variables, and a nonlinear version for continuous variables but without modularity regularization (i.e. only $TC(Z) + TC(X|Z)$) was used in [41].

## 6 Conclusion

By characterizing a class of structured latent factor models via an information-theoretic criterion, we were able to design a new approach for structure learning that outperformed standard approaches while also reducing stepwise computational complexity from cubic to linear. Better scaling allows us to apply our approach to very high-dimensional data like full-resolution fMRI, recovering biologically plausible structure thanks to our inductive prior on modular structure. A bias towards modular latent factors may not be appropriate in all domains and, unlike methods encoding sparsity priors (e.g., GLASSO), our approach leads to a non-convex optimization and therefore no theoretical guarantees. Nevertheless, we demonstrated applicability across a diverse set of over fifty real-world datasets, with especially promising results in domains like gene expression and finance where we outperform sparsity-based methods by large margins both in solution quality and computational cost.

### Acknowledgments

We thank Andrey Lokhov, Marc Vuffray, and Seyoung Yun for valuable conversations about this work and we thank anonymous reviews whose comments have greatly improved this manuscript. H. Harutyunyan is supported by USC Annenberg Fellowship. This work is supported in part by DARPA via W911NF-16-1-0575 and W911NF-17-C-0011, and the Office of the Director of National Intelligence (ODNI), Intelligence Advanced Research Projects Activity (IARPA), via 2016-16041100004. The views and conclusions contained herein are those of the authors and should not be interpreted as necessarily representing-the official policies, either expressed or implied, of DARPA, ODNI, IARPA, or the U.S. Government. The U.S. Government is authorized to reproduce and distribute reprints for governmental purposes notwithstanding any copyright annotation therein.

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
