[Supplementary Material]

# Supplementary material: fast structure learning with modular regularization

## A  Proofs

### A.1  Proof of proposition 2.1

**Proposition. 2.1 restated.** The random variables $X$ and $Z$ are described by a directed graphical model where the parents of $X$ are in $Z$ and the $Z$'s are independent if and only if $TC(X|Z) + TC(Z) = 0$.

*Proof.* Because $TC$ is always non-negative,

$$TC(X|Z) + TC(Z) = 0 \Leftrightarrow TC(Z) = 0 \text{ and } TC(X|Z) = 0.$$

We also have the following standard statements [6]

$$TC(X|Z) = 0 \Leftrightarrow \forall x, z, \; p(x|z) = \prod_{i=1}^{p} p(x_i|z),$$

$$TC(Z) = 0 \Leftrightarrow \forall z, \; p(z) = \prod_{j=1}^{m} p(z_j).$$

Putting these together, we have

$$\forall x, z, \; p(x, z) = \prod_{i=1}^{p} \prod_{j=1}^{m} p(x_i|z) p(z_j).$$

We can see that this statement is equivalent to the definition of a Bayesian network for random variables $X, Z$ with respect to the graph in Fig. 1a. □

### A.2  Proof of theorem 2.1

**Theorem. 2.1 restated.** A multivariate Gaussian distribution $p(x, z)$ is a modular latent factor model if and only if $TC(X|Z) + TC(Z) = 0$ and $\forall i, TC(Z|X_i) = 0$.

*Proof.* First we show that for any modular latent factor model, even non-Gaussian, the constraints are satisfied. Thm. 2.1 establishes that the model implies $TC(X|Z) + TC(Z) = 0$. We must show that the additional restriction that each $X_i$ has only one parent, $Z_{\pi_i}$, implies the condition $\forall i, TC(Z|X_i) = 0$. Looking at the rules for d-separation we see that $Z_1, \ldots, Z_m$ are independent conditioned on $X_i$. Therefore, $\forall i, TC(Z|X_i) = 0$.

Now, we show that a multivariate Gaussian distribution $p(x, z)$ with $TC(X|Z) + TC(Z) = 0$ and $\forall i, TC(Z|X_i) = 0$ is a modular latent factor model:

$$\forall x, z, \; p(x, z) = \prod_{i=1}^{p} p(x_i|z_{\pi_i}) \prod_{j=1}^{m} p(z_j), \text{ for some } \pi_i \in \{1, 2, \ldots, m\}.$$

By Thm. 2.1 we have that $\forall x, z, \; p(x, z) = \prod_{i=1}^{p} p(x_i|z) \prod_{j=1}^{m} p(z_j)$. To complete the proof we show that $(TC(Z) = 0 \;\&\; TC(Z|X_i) = 0) \Rightarrow p(x_i|z) = p(x_i|z_{\pi_i})$ for some $\pi_i \in \{1, \ldots, m\}$. We have

$$p(x_i|z) = p(x_i)/p(z) \prod_{j=1}^{m} p(z_j|x_i)$$

$$= p(x_i) \prod_{j=1}^{m} p(z_j|x_i)/p(z_j)$$

$$= p(x_i) \prod_{j=1}^{m} p(z_j, x_i)/(p(x_i)p(z_j)). \tag{5}$$

We also have that $TC(Z|X_i) = 0 \Rightarrow \forall j \neq k, \mathrm{Cov}[Z_j, Z_k|X_i] = 0$. For Gaussians $\mathrm{Cov}[Z_j, Z_k|X_i] = \mathrm{Cov}[Z_j, Z_k] - \mathrm{Cov}[Z_j, X_i]\mathrm{Cov}[Z_k, X_i]/\mathrm{Var}[X_i]$. Having $\mathrm{Var}[X_i] > 0$ and $(TC(Z) = 0 \Rightarrow \mathrm{Cov}[Z_j, Z_k] = 0)$, we get $\mathrm{Cov}[Z_j, X_i] = 0 \vee \mathrm{Cov}[Z_k, X_i] = 0$. Therefore, for all but at most one index, $\pi_i$, it must be the covariance of $X_i$ and $Z_j$ is zero, so that $p(z_j, x_i) = p(x_i)p(z_j)$. Putting this in Eq. (5) we get $p(x_i|z) = p(x_i|z_{\pi_i})$.

Note that we cannot remove the Gaussian assumption, since it is possible to have $TC(X|Z) = 0, TC(Z) = 0$, and $\forall i, TC(Z|X_i) = 0$, but still have two non-trivial parents for one $X_i$. For example, if $Z_1, Z_2 \stackrel{iid}{\sim} \mathrm{Bernoulli}(^1/_2)$ and $X_1 = 2Z_1 + Z_2$. It can be easily seen that the conditions are satisfied, but it is impossible to model $X_1$ with only $Z_1$ or $Z_2$ as its parent. □

# B  Complete derivation of linear CorEx

In this section we describe the complete derivation of linear CorEx. The first step is to define the family of joint distributions we are searching over by parametrizing $p_W(z|x)$. If $X_{1:p}$ is Gaussian, then we can ensure $X_{1:p}, Z_{1:m}$ are jointly Gaussian by parametrizing $p_W(z_j|x) = \mathcal{N}(w_j^T x, \eta_j^2)$, $w_j \in \mathbb{R}^p, j = 1..m$, or equivalently by $z = Wx + \epsilon$ with $W \in \mathbb{R}^{m \times p}, \epsilon \sim \mathcal{N}(0, \mathrm{diag}(\eta_1^2, \ldots, \eta_m^2))$. The noise variances $\eta_j^2$ are taken to be constants. Please note the implicit conditional independence assumption, $TC(Z|X) = 0$, we are making using this parameterization. We do this assumption since modular latent factor models have $TC(Z|X) = 0$, and it simplifies further derivations. W.l.o.g. we assume the data is standardized so that $\mathbb{E}[X_i] = 0, \mathbb{E}[X_i^2] = 1$.[1] If it is not standardized we can standardize it using the empirical means and standard deviations. Motivated by Thm. 2.1, we will start with the following optimization problem:

$$\underset{W}{\text{minimize}}\, TC(X|Z) + TC(Z) + \sum_{i=1}^{p} Q_i, \qquad (6)$$

where $Q_i$ are regularization terms for encouraging modular solutions (i.e. encouraging solutions with smaller value of $TC(Z|X_i)$.[2] We will later specify this regularizer as a non-negative quantity that goes to zero in the case of exactly modular latent factor models. The $TC(X|Z) + TC(Z)$ part of the Eq. (6) can be rewritten as follows:

$$
\begin{aligned}
TC(X|Z) + TC(Z) &= \sum_{i=1}^{p} H(X_i|Z) - H(X|Z) + \sum_{j=1}^{m} H(Z_j) - H(Z) \\
&= \sum_{i=1}^{p} H(X_i|Z) + \sum_{j=1}^{m} H(Z_j) - (H(X|Z) + H(Z)) \\
&= \sum_{i=1}^{p} H(X_i|Z) + \sum_{j=1}^{m} H(Z_j) - (H(Z|X) + H(X)) \\
&= \sum_{i=1}^{p} H(X_i|Z) + \sum_{j=1}^{m} (H(Z_j) - H(Z_j|X)) + H(X) \\
&\propto \sum_{i=1}^{p} H(X_i|Z) + \sum_{j=1}^{m} I(Z_j; X). \qquad (7)
\end{aligned}
$$

The first two lines invoke definitions and re-arrange. The third line uses Bayes' rule to rewrite the entropies. The fourth line invokes conditional independence of $Z$'s conditioned on X. Next, we write

out the explicit form of expressions in Eq. (7) for Gaussians and ignore constants:

$$
\begin{aligned}
\sum_{i=1}^{p} H(X_i|Z) &+ \sum_{j=1}^{m} I(Z_j; X) \\
&= \sum_{i=1}^{p} \frac{1}{2} \mathbb{E}_Z \log \left(2\pi e \mathrm{Var}[X_i|Z]\right) + \sum_{j=1}^{m} \left(H(Z_j) - H(Z_j|X)\right) \\
&= \sum_{i=1}^{p} \frac{1}{2} \log \mathbb{E}_Z \left[2\pi e \mathrm{Var}[X_i|Z]\right] + \sum_{j=1}^{m} \left(H(Z_j) - H(Z_j|X)\right) \\
&\propto \frac{1}{2} \sum_{i=1}^{p} \log \mathbb{E}\left[(X_i - \mathbb{E}_{X_i|Z}[X_i|Z])^2\right] + \frac{1}{2} \sum_{j=1}^{m} \left(\log \mathrm{Var}[Z_j] - \mathbb{E}_X \log \mathrm{Var}[Z_j|X]\right) \\
&\propto \frac{1}{2} \sum_{i=1}^{p} \log \mathbb{E}\left[(X_i - \mathbb{E}_{X_i|Z}[X_i|Z])^2\right] + \frac{1}{2} \sum_{j=1}^{m} \left(\log \mathbb{E}\left[Z_j^2\right] - \log(\eta_j^2)\right) \\
&\propto \frac{1}{2} \sum_{i=1}^{p} \log \mathbb{E}\left[(X_i - \mathbb{E}_{X_i|Z}[X_i|Z])^2\right] + \frac{1}{2} \sum_{j=1}^{m} \log \mathbb{E}\left[Z_j^2\right].
\end{aligned}
\tag{8}
$$

We used the fact that the differential entropy of a Gaussian variable with variance $\sigma^2$ is equal to $1/2 \log(2\pi e \sigma^2)$. Also, we used the fact that if $A, B$ are jointly Gaussian random variables, then $H(A|B) \propto \mathbb{E}_B \log \mathrm{Var}[A|B] = \log \mathbb{E}_B \mathrm{Var}[A|B]$. The logarithm and expectation can be swapped because for Gaussians $\mathrm{Var}[A|B]$ is constant for any value of $B$. In the fifth line we replace $\mathrm{Var}[Z_j]$ with $\mathbb{E}\left[Z_j^2\right]$, because having $\mathbb{E}[X] = 0$ and $z_j = w_j^T x + \epsilon_j$ implies $\mathbb{E}[Z_j] = 0$. Considering Eq. (8), the problem (6) becomes:

$$
\underset{W}{\text{minimize}} \sum_{i=1}^{p} \left(1/2 \log \mathbb{E}\left[(X_i - \mu_{X_i|Z})^2\right] + Q_i\right) + \sum_{j=1}^{m} 1/2 \log \mathbb{E}\left[Z_j^2\right],
\tag{9}
$$

where $\mu_{X_i|Z} = \mathbb{E}_{X_i|Z}[X_i|Z]$. For Gaussians, calculating $\mu_{X_i|Z}$ requires a computationally undesirable matrix inversion. Instead, we will select $Q_i$ to eliminate this term while also encouraging modular structure. According to Thm. 2.1, modular models obey $TC(Z|X_i) = 0$, which implies that $p(x_i|z) = p(x_i)/p(z) \prod_j p(z_j|x_i)$. Let $\nu_{X_i|Z}$ be the conditional mean of $X_i$ given $Z$ under such factorization. Then it will have the following form (see Sec. B.1 for the derivation):

$$
\nu_{X_i|Z} = \frac{1}{1 + r_i} \sum_{j=1}^{m} \frac{Z_j B_{j,i}}{\sqrt{\mathbb{E}\left[Z_j^2\right]}},
$$

$$
\text{with } R_{j,i} = \frac{\mathbb{E}[X_i Z_j]}{\sqrt{\mathbb{E}\left[X_i^2\right] \mathbb{E}\left[Z_j^2\right]}}, B_{j,i} = \frac{R_{j,i}}{1 - R_{j,i}^2}, r_i = \sum_{j=1}^{m} R_{j,i} B_{j,i}.
$$

We see that computing $\nu_{X_i|Z}$ is easier since it requires no matrix inversion and depends only on pairwise statistics between observed and latent variables. If we let $Q_i = 1/2 \log \mathbb{E}\left[(X_i - \nu_{X_i|Z})^2\right] - 1/2 \log \mathbb{E}\left[(X_i - \mu_{X_i|Z})^2\right]$, we will replace $\mu_{X_i|Z}$ with $\nu_{X_i|Z}$ in problem (9). To see why this also

encourages modular structures we note that

$$Q_i = \frac{1}{2} \log \mathbb{E}\left[(X_i - \nu_{X_i|Z})^2\right] - \frac{1}{2} \log \mathbb{E}\left[(X_i - \mu_{X_i|Z})^2\right]$$

$$= \frac{1}{2} \log \frac{\mathbb{E}\left[(X_i - \nu_{X_i|Z})^2\right]}{\mathbb{E}\left[(X_i - \mu_{X_i|Z})^2\right]}$$

$$= \frac{1}{2} \log \left( \frac{\mathbb{E}\left[(X_i - \nu_{X_i|Z} + \mu_{X_i|Z} - \mu_{X_i|Z})^2\right]}{\mathbb{E}\left[(X_i - \mu_{X_i|Z})^2\right]} \right)$$

$$= \frac{1}{2} \log \left( \frac{\mathbb{E}\left[(X_i - \mu_{X_i|Z})^2\right] + \mathbb{E}\left[(\mu_{X_i|Z} - \nu_{X_i|Z})^2\right] + 2\mathbb{E}\left[(X_i - \mu_{X_i|Z})(\mu_{X_i|Z} - \nu_{X_i|Z})\right]}{\mathbb{E}\left[(X_i - \mu_{X_i|Z})^2\right]} \right)$$

$$= \frac{1}{2} \log \left( 1 + \frac{\mathbb{E}\left[(\mu_{X_i|Z} - \nu_{X_i|Z})^2\right] + 2\mathbb{E}_Z \mathbb{E}_{X_i|Z}\left[(X_i - \mu_{X_i|Z})(\mu_{X_i|Z} - \nu_{X_i|Z})\right]}{\mathbb{E}\left[(X_i - \mu_{X_i|Z})^2\right]} \right)$$

$$= \frac{1}{2} \log \left( 1 + \frac{\mathbb{E}\left[(\mu_{X_i|Z} - \nu_{X_i|Z})^2\right]}{\mathbb{E}\left[(X_i - \mu_{X_i|Z})^2\right]} \right) \geq 0.$$

We see that this regularizer is always non-negative and is zero exactly for modular latent factor models (when $\mu_{X_i|Z} = \nu_{X_i|Z}$). Summing up, the final objective simplifies to the following:

$$\underset{W}{\text{minimize}} \sum_{i=1}^{p} \nicefrac{1}{2} \log \mathbb{E}\left[(X_i - \nu_{X_i|Z})^2\right] + \sum_{j=1}^{m} \nicefrac{1}{2} \log \mathbb{E}\left[Z_j^2\right]. \tag{10}$$

This objective depends on pairwise statistics and requires no matrix inversion. The global minimum is achieved for modular latent factor models. The next step is to approximate the expectations in the objective (3) with empirical means and optimize it with respect to the parameters $W$.

After training the method we can interpret $\hat{\pi}_i \in \arg\max_j I(Z_j; X_i) = \arg\max_j -\nicefrac{1}{2} \log(1 - R_{j,i}^2) = \arg\max_j |R_{j,i}|$ as the parent of variable $X_i$. Additionally, we can estimate the covariance matrix of the observed variables. The method we use for estimating the covariance is as follows. First, we have assumed that the data is standardized, so we just need to calculate the off-diagonal terms. If $TC(X|Z) = 0$, this implies the conditional covariance of $X$ given $Z$ is diagonal. Additionally, using the law of total covariance we have:

$$\text{Cov}\left[X_i, X_{\ell \neq i}\right] = \mathbb{E}\left[\text{Cov}[X_i, X_\ell|Z]\right] + \text{Cov}\left[\mu_{X_i|Z}, \mu_{X_\ell|Z}\right].$$

By combining the last two statements we get:

$$\mathbb{E}\left[\text{Cov}[X_i, X_{\ell \neq i}|Z]\right] = \mathbb{E}\left[X_i X_\ell\right] - \mathbb{E}\left[\mu_{X_i|Z} \mu_{X_\ell|Z}\right] = 0.$$

If we assume the constraints $TC(Z) = 0$ & $\forall i, TC(Z|X_i) = 0$ are satisfied, we saw that this implies $\mu_{X_i|Z} = \nu_{X_i|Z}$. Also, as $TC(Z) = 0 \Rightarrow \mathbb{E}[Z_j Z_k] = \delta_{j,k} \mathbb{E}\left[Z_j^2\right]$, the off-diagonal elements of $\mathbb{E}\left[X_i X_\ell\right]$ satisfy:

$$\mathbb{E}\left[X_i X_{\ell \neq i}\right] = \mathbb{E}\left[\nu_{X_i|Z} \nu_{X_\ell|Z}\right] = \frac{(B^\top B)_{i,\ell}}{(1 + r_i)(1 + r_\ell)}.$$

In conclusion we get the following covariance matrix estimates:

$$\widehat{\Sigma}_{i,\ell \neq i} = \frac{(B^T B)_{i,\ell}}{(1 + r_i)(1 + r_\ell)}, \quad \widehat{\Sigma}_{i,i} = 1. \tag{11}$$

Note that the covariance matrix estimate corresponds to the covariance matrix of the learned model if $TC(X|Z) = 0, TC(Z) = 0$, and $\forall i, TC(Z|X_i) = 0$, i.e. the learned model is modular. Otherwise it is an approximation of to the covariance matrix of the learned model. From Eq. 11 we see that the estimates are low-rank plus diagonal matrices. In case when the learned model is modular, it is also block-diagonal with each block being a diagonal plus rank-one matrix. Therefore, encouraging modular structures pushes the low-rank covariance estimate to be also block-diagonal with each block being a diagonal plus rank-one matrix.

## B.1 Derivation of the conditional mean under modularity constraints

Under the conditions that $X, Z$ are jointly Gaussian and $\forall i, TC(X|Z_i) = 0$, we would like to derive the mean of $X_i$ conditioned on $Z$, $\nu_{X_i|Z}$. We have that $TC(Z|X_i) = 0 \Rightarrow p(x_i|z) = p(x_i)/p(z)\prod_j p(z_j|x_i)$. We will look at the distribution $q(x_i|z) = p(x_i)/p(z)\prod_j p(z_j|x_i)$ and calculate the conditional mean of this distribution.

Let $R_{j,i}$ be the Pearson correlation coefficient between $Z_j$ and $X_i$ whose means and standard deviations are respectively indicated with $\nu_j, \rho_j$ and $\mu_i, \sigma_i$ (all with respect to the distribution $p$). The marginal distribution for the Gaussian distribution relating $Z_j$ and $X_i$ is well known:

$$p(z_j|x_i) = \mathcal{N}(\nu_j + R_{j,i}\rho_j/\sigma_i(x_i - \mu_i), (1 - R_{j,i}^2)\rho_j^2).$$

Now we look only at the exponents of $q(x_i|z)$, ignoring the normalization, to get the following:

$$-\log q(x_i|z) \propto (x_i - \mu_i)^2/\sigma_i^2 + \sum_{j=1}^{m}(z_j - \nu_j - R_{j,i}\rho_j/\sigma_i(x_i - \mu_i))^2/((1 - R_{j,i}^2)\rho_j^2).$$

Collecting only the terms involving $x_i$ we get the following:

$$-\log q(x_i|z) \propto Ax_i^2 + Bx_i + C,$$

$$\text{with } A = 1/\sigma_i^2 + \sum_{j=1}^{m} \frac{R_{j,i}^2\rho_j^2/\sigma_i^2}{(1 - R_{j,i}^2)\rho_j^2}, \ B = -2\mu_i/\sigma_i^2 - \sum_{j=1}^{m} \frac{2(z_j - \nu_j + \mu_i R_{j,i}\rho_j/\sigma_i)R_{j,i}\rho_j/\sigma_i}{(1 - R_{j,i}^2)\rho_j^2}.$$

From completing the square, we see that the conditional mean of $X_i|Z$ has the form $\nu_{X_i|Z} = -B/(2A)$.

Finally, we simplify the formulae because $\mu_i = \mathbb{E}[X_i] = \nu_j = \mathbb{E}[Z_j] = 0$ and $\sigma_i^2 = \mathbb{E}[X_i^2] = 1$. This implies that $R_{j,i} = \mathbb{E}[X_iZ_j]/\sqrt{\mathbb{E}[X_i^2]\mathbb{E}[Z_j^2]}$, leaving us with the following form:

$$\nu_{X_i|Z} = \frac{1}{1 + r_i}\sum_{j=1}^{m} B_{j,i}\frac{Z_j}{\sqrt{\mathbb{E}[Z_j^2]}}, \ \text{ with } B_{j,i} = \frac{R_{j,i}}{1 - R_{j,i}^2}, r_i = \sum_{j=1}^{m} R_{j,i}B_{j,i}.$$

## C  Sample complexity lower bound

In this section we derive a lower bound on sample complexity for learning the structure of modular latent factor model. We follow the construction of information-theoretic sample complexity bounds in [7].

**Theorem C.1.** For a multivariate Gaussian modular latent factor model with $p$ observed variables $X_{1:p}$, $m$ latent variables $Z_{1:m}$ with $p/m$ children each and additive white Gaussian noise channel from parent to child with signal-to-noise ratio $s$, the number of samples, $n$, required to recover the structure of the graphical model with error probability $\epsilon$ is lower bounded as

$$n \geq \frac{2\left((1 - \epsilon)\log\left(\binom{p}{p/m,\ldots,p/m}\frac{1}{m!}\right) - 1\right)}{(p - 1)\log(1 + s\frac{1 - 1/m}{1 - 1/p}) - (m - 1)\log(1 + s\frac{p}{m})}. \tag{12}$$

*Proof.* Consider the class of modular latent factor models with $p$ observed variables and $m$ latent factors each having exactly $p/m$ children. To distinguish the structure among this class of models corresponds to partitioning the observed variables into $m$ equally sized groups. The number of such groupings is,

$$M = \binom{p}{p/m,\ldots,p/m}\frac{1}{m!},$$

the multinomial coefficient for dividing $p$ items into $m$ equally sized boxes, divided by the number of indistinguishable permutations among boxes, $m!$. We take $\theta \in \{1,\ldots,M\}$ to be an index specifying a model in this ensemble. Now learning the structure corresponds to finding $\theta$ from data.

W.l.o.g. assume $\forall j, \text{Var}[Z_j] = b$. Then $X_i = Z_{\pi_\theta(i)} + \eta_i$, where $\pi_\theta(i)$ is the index of the parent of $X_i$ in model $\theta$ and $\eta_i$ is independent Gaussian noise with variance $a$. Since we have fixed the signal-to-noise ratio, we have that $a = b/s$. W.l.o.g. we can assume that $\forall i, \mathbb{E}[X_i] = 0$. Then the covariance matrix of observed variables, $\Sigma_{\theta,i,j} = \mathbb{E}[X_i X_j] = b\delta_{\pi_\theta(i),\pi_\theta(j)} + a\delta_{i,j}$, where $\delta$ is the Kronecker delta.

Fano's inequality tells us that the probability of an error, $\epsilon$, in picking the correct index $\theta$ given $n$ samples of data, $X_{1:p}^{1:n}$, is bounded as follows:

$$\epsilon \geq 1 - \frac{I(\theta; X_{1:p}^{1:n}) + 1}{\log M}.$$

Following [7], we use an upper bound for the mutual information, $I(\theta; X_{1:p}^{1:n}) \leq nF/2$, where

$$F = \log \det \bar{\Sigma} - 1/M \sum_{\theta=1}^{M} \log \det \Sigma_\theta,$$

and $\bar{\Sigma} = 1/M \sum_{\theta=1}^{M} \Sigma_\theta$. Re-arranging Fano's inequality gives the following sample complexity bound:

$$n \geq 2\frac{(1 - \epsilon)\log M - 1}{F}.$$

All that remains is to find an expression for $F$. To build intuition, we explicitly write out the case for $p = 4, m = 2$, and for some $\theta$.

$$\Sigma_\theta = \begin{bmatrix} b+a & b & 0 & 0 \\ b & b+a & 0 & 0 \\ 0 & 0 & b+a & b \\ 0 & 0 & b & b+a \end{bmatrix}$$

Clearly this is a block diagonal matrix where each block is a diagonal plus rank-one (DPR1) matrix. After we average over all $\theta$ to get $\bar{\Sigma}$, every off-diagonal entry will be the same, equal to the probability of $j \neq i$ being in the same group as $i$, or $(p/m - 1)/(p - 1)$. Therefore $\bar{\Sigma}$ is also a DPR1 matrix. Using standard identities for block diagonal and DPR1 matrices, we calculate the determinants:

$$\det \Sigma_\theta = a^p \left(1 + \frac{b}{a}\frac{p}{m}\right)^m,$$

$$\det \bar{\Sigma} = a^p \left(1 + \frac{b}{a}\frac{p}{m}\right)\left(1 + \frac{b}{a}\frac{p}{m}\left(\frac{m-1}{p-1}\right)\right)^{p-1}.$$

Finally, we can combine all of these expressions to get a lower bound for sample complexity that depends only on $p, m$, and the signal-to-noise ratio, $s = b/a$. $\qquad\square$

The bound of Thm. C.1 is not very intuitive because it involves logarithm of a multinomial coefficient. We provide a simpler asymptotic expression for the bound. Using Stirling's approximation we have that $\log \binom{p}{p/m,...,p/m}\frac{1}{m!} \approx p\log m + 1/2\log(p/m) - m/2\log(m\, p\, 2\pi/e^2)$ for large values of $p$. In the limit of large $p$, this approximation gives us the following lower bound:

$$n \geq \frac{2(1 - \epsilon)\log m}{\log\left(1 + s(1 - 1/m)\right)}.$$

Wee see that in the limit of large $p$ the bound becomes constant rather than becoming infinite. Moreover, when we plot the lower bound of Eq. (12) in Fig. 7, we see that for fixed number of latent factors the bound goes down as we increase $p$. These two facts together hint (but do not prove) that modular latent factor models may allow blessing of dimensionality. An evidence of blessing of dimensionality is demonstrated in Sec. 4. Intuitively, recovery gets easier because more variables provide more signal to reconstruct the fixed number of latent factors. While it is tempting to retrospectively see this as obvious, the same argument could be (mistakenly) applied to other families of latent factor models, such as the unconstrained latent factor models shown in Fig. 1a, for which the sample complexity grows as we increase $p$ [2, 15].

Figure 7: Theorem C.1 prevents perfect structure recovery in the shaded region. On the left: for fixed signal-to-noise ratio and number of latent factors, the lower bound of Thm. C.1 decreases as the number of observed variables increases. On the right: the same effect is visible for other values of signal-to-noise ratio.

## D   Implementation details

In this section we present details on baselines, experiments, and hyperparameters.

**Baselines**   For factor analysis, PCA, sparse PCA, independent component analysis, k-means clustering, spectral clustering, negative matrix factorization, hierarchical agglomerative clustering using Euclidean distance, hierarchical agglomerative clustering the Ward linkage rule, Ledoit-Wolf and graphical LASSO we used the scikit-learn implementations [42]. We implemented latent tree modeling with the "Relaxed RG" method. We slightly modified latent tree modeling to use the same prior information as other methods in the comparison, namely, that there are exactly m groups and observed nodes can be siblings, but not parent and child. For latent variable graphical LASSO, we used the implementation available in the REGAIN repository.[3]

**Experimental setups**   In the blessing of dimensionality experiments all methods were given the correct number of clusters. The scores were computed using 10000 test examples. When possible we reported the means and standard deviations over 20 runs. In the covariance estimation experiments with synthetic data, models requiring a number of latent factors or a number of components were given the correct number. The scores were computed using 1000 test examples. We reported the means and standard deviations over 5 runs. In the stock market experiments models were trained on $n$ weeks and their estimates were evaluated using the negative log-likelihood on the subsequent 26 weeks. We presented the average score from rolling the training and testing sets over the entire time period. Standard deviations are not presented because scores corresponding to different time periods are very different, resulting in large standard deviations. This is due to the stock market exhibiting different behaviour in different time periods. In experiments with OpenML datasets we used a random 80-20 train-test split. We reported the negative log-likelihood on test sets. As large amount of computation is needed to generate results on OpenML datasets, we did only a single run for each dataset.

**Hyperparameters**   In all cases the proposed method was trained using Adam optimizer with $0.01$ learning rate, $\beta_1 = 0.9$, and $\beta_2 = 0.999$. In all covariance estimation problems the hyperparameters were selected from a grid of values using a 3-fold cross-validation procedure. The sparsity parameter of sparse PCA was selected from $[0.1, 0.3, 1.0, 3.0, 10.0]$. The sparsity parameters of GLASSO and latent variable GLASSO were selected from $[0.01, 0.1, 0.3, 1.0, 3.0, 10.0]$. For latent variable GLASSO, the additional regularization parameter ("tau", controlling the nuclear norm of the low-rank part of the inverse covariance matrix) was selected from $[0.01, 0.1, 1.0, 10.0, 100.0]$. In the experiments with OpenML datasets the sparsity hyperparameter of BigQUIC was selected form $[2^0, 2^1, 2^2, 2^3]$. In the timing experiments the sparsity parameters of sparse PCA and GLASSO were set to $1.0$. LVGLASSO was trained with the sparsity parameter set to $0.1$ and with "tau" set to $30.0$.

(a) Modular latent factor model

(b) Modular + extra parents

(c) Modular + correlated $Z$s

(d) Modular + extra parents and correlated $Z$s

Figure 8: Empirical covariance matrices (estimated using $n = 10^4$ samples) corresponding to modular (a) and approximately modular (b, c, d) latent factor models. In all examples $m = 8$, $p = 128$, $s = 0.5$.

## E   Details on generating synthetic data

In all experiments involving a synthetic modular latent factor model we generate the data the following way. We first take $m$ independent standard Gaussian random variables, $Z_1, Z_2, \ldots, Z_m \overset{iid}{\sim} \mathcal{N}(0, 1)$. For simplicity we assume that $m$ divides $p$ and each latent factor has exactly $p/m$ children. W.l.o.g. we connect the first $p/m$ observed variables with $Z_1$, then next $p/m$ variables with $Z_2$ and so on. We assume additive white Gaussian noise channel with signal-to-noise ratio $s$ from each parent to its children. In this setup, we set $X_i = \sqrt{\frac{s}{s+1}} Z_{\pi_i} + \sqrt{\frac{1}{s+1}} \eta_i$, where $\pi_i$ is the index of the parent of $X_i$, and $\eta_i$ is independent standard Gaussian noise. Fig. 8a shows a covariance matrix corresponding to a modular latent factor models created using the described procedure.

To create approximately modular latent factor models we do two modifications on a modular latent factor model: correlating the latent variables and adding extra parents for observed variables. For correlating the latent factors we take $m$ independent standard normal random variables $\xi_j, j = 1..m$ and compute $z_j = (\sqrt{2}\xi_j + \xi_u + \xi_v)/2$, where $u, v \sim \text{Uniform}\{1, 2, \ldots, m\}$. For adding extra parents, we randomly sample $p$ extra edges from a latent factor to a non-child observed variable. By this we create on average one extra edge per each observed variable. To keep the notion of clusters well-defined, we make sure that each observed variable has higher mutual information with its main parent compared to that with added extra parents. Suppose some $X_i$ has $k$ extra parents, $Z_{\tau_1}, \ldots, Z_{\tau_k}$. Then we splits $\frac{s}{s+1}$ – the variance of the signal in a pure modular case – into $k + 2$ equal parts,

(a) Modular latent factor model

(b) Modular + extra parents

(c) Modular + correlated $Z$s

(d) Modular + extra parents and correlated $Z$s

Figure 9: Mutual information matrices between observed variables and latent factors linear CorEx produces when it is trained on a modular (a) and approximately modular (b, c, d) latent factor models. In all examples $m = 8$, $p = 128$, $s = 5.0$.

Figure 10: Mutual information matrix between observed variables (stocks) and latent factors linear CorEx produces when trained on the stock marked data (January 2014-January 2017).

$\delta = \frac{s}{(s+1)(k+2)}$. We then set $X_i = \sqrt{2\delta}Z_{\pi_i} + \sqrt{\delta}Z_{\tau_1} + \cdots + \sqrt{\delta}Z_{\tau_k} + \sqrt{\frac{1}{s+1}}\eta_i$, where again $\eta_i$ is independent standard Gaussian noise. Figs. 8b, 8c, 8d show covariance matrices corresponding to approximately modular latent factor models created using the described procedures.

# F   Additional results

In this section we provide additional results that were not presented in the main text due to the space constraints.

## F.1   Examining the modularity of learned models

We do visualizations to see whether the regularization term of linear CorEx actually leads to learning modular (or approximately modular) latent factor models. We examine the mutual information matrices between observed and latent variables that linear CorEx produces when it is trained on different types of synthetic data (see Fig 9). We see that the regularization term we add for encouraging modular structures is indeed effective.

Figure 11: Inverse covariance matrix of some of the S&P 500 stocks. Plotted are the cells that have absolute value greater than $0.015$.

Next, we look at the same mutual information matrix for stock market data. Fig. 10 shows the mutual information matrix for S&P 500 stocks belonging to "consumer discretionary", "consumer staples", "energy", "financials", and "health care" sectors. We see that most of the stocks have significant mutual information only with a few latent factors. Moreover, stocks belonging to the same sector are likely to share a parent. Additionally, we visualize the inverse covariance matrix of these stocks (see Fig. 11). For Gaussian random variables the thresholded inverse covariance matrix can be interpreted as a random Markov field. We see that it is almost block-diagonal, but has some off-diagonal connections, confirming that the learned model is close to being a modular latent factor model.

Summing up, all these visualizations assert that the linear CorEx succeeds in biasing the model selection procedure towards modular structures. More importantly, we see that when the pure modular structure is inappropriate, it picks solutions that are close to being modular.

### F.2 Results on OpenML datasets

Table 2 presents a comparison of various covariance estimation baselines on 51 OpenML datasets.

Table 2: This table compares covariance estimates on OpenML data. Scores reported are negative log-likelihood (lower better) and the best entry is bolded. Scores orders of magnitude larger than the best score or evaluating to NaN are shortened with "*". Methods compared, in order, are PCA, Sparse PCA, factor analysis, Ledoit-Wolf, GLASSO (using the BigQUIC algorithm), and the method proposed in this paper.

| ID:Dataset | p | n | Methods | | | | | |
|---|---|---|---|---|---|---|---|---|
| | | | PCA | SPCA | FA | LW | BigQUIC | Proposed |
| 5:arrhythmia | 206 | 54 | 178 | -33 | * | 164 | * | **-74** |
| 407:krystek | 1143 | 24 | 2122 | -1748 | * | 707 | -1428 | **-2816** |
| 408:depreux | 1143 | 20 | 1454 | * | * | 852 | * | **-482** |
| 409:pdgfr | 321 | 63 | 112 | 40 | * | 83 | 364 | **-6** |
| 410:carbolenes | 1143 | 29 | 1900 | * | * | 923 | * | **-8** |
| 419:PHENETYL1 | 629 | 17 | 876 | 560 | 5584 | **281** | 1041 | 286 |
| 420:cristalli | 1143 | 25 | 1846 | * | * | 1366 | * | **780** |
| 424:pah | 113 | 64 | -129 | 47 | **-188** | 25 | 134 | 58 |
| 439:chang | 1143 | 27 | 2331 | * | * | 1058 | * | **-6** |
| 1017:arrhythmia | 206 | 54 | 178 | -33 | * | 164 | * | **-60** |
| 1104:leukemia | 7129 | 57 | 17028 | 13164 | 396636 | **7019** | 11530 | 7336 |
| 1107:tumorsC | 7129 | 48 | 16990 | 8499 | 9642 | **8070** | 9398 | 8399 |
| 1122:APBreastProstat | 10935 | 330 | 18427 | 17219 | 17741 | 13431 | 17002 | **10639** |
| 1123:APEndometriumBr | 10935 | 324 | 18960 | 12616 | 12720 | 11356 | 18330 | **10452** |
| 1124:APOmentumUterus | 10935 | 160 | 84928 | 82496 | 82656 | 66784 | 76832 | **66176** |
| 1125:APOmentumProsta | 10935 | 116 | 90024 | 100392 | 100032 | 69168 | 81264 | **67560** |
| 1126:APColonLung | 10935 | 329 | 84612 | 76362 | 76626 | **66198** | 76098 | 67188 |
| 1127:APBreastOmentum | 10935 | 336 | 86020 | 88196 | 88604 | **66824** | 79968 | 68408 |
| 1128:OVABreast | 10935 | 1236 | 83626 | 79434 | 79087 | **64951** | 76483 | 70308 |
| 1129:APUterusKidney | 10935 | 307 | 85498 | 74276 | 74214 | **68882** | 75764 | **68882** |
| 1130:OVALung | 10935 | 1236 | 81989 | * | 73904 | 81518 | 76409 | **69291** |
| 1131:APProstateUteru | 10935 | 154 | 85653 | 73687 | 73625 | **67208** | 74617 | 67363 |
| 1132:APOmentumLung | 10935 | 162 | 88803 | 83820 | 83424 | 70917 | 79002 | **68805** |
| 1133:APEndometriumCo | 10935 | 277 | 84840 | 75376 | 75152 | **65576** | 76776 | 66920 |
| 1134:OVAKidney | 10935 | 1236 | 81964 | 75144 | 73507 | 81592 | 76210 | **69242** |
| 1135:APColonProstate | 10935 | 284 | 84702 | 82365 | 82194 | **65550** | 78318 | 67260 |
| 1136:APLungUterus | 10935 | 200 | 87200 | 77880 | 77360 | 69200 | 76480 | **68440** |
| 1137:APColonKidney | 10935 | 436 | 83776 | 73515 | 73084 | **66616** | 75882 | 68094 |
| 1138:OVAUterus | 10935 | 1236 | 81964 | 74772 | 73358 | 81493 | 76136 | **69242** |
| 1139:OVAOmentum | 10935 | 1236 | 81964 | 75442 | 74152 | 81567 | 76458 | **69266** |
| 1140:APOvaryLung | 10935 | 259 | 87464 | * | 85280 | 86320 | 79560 | **68380** |
| 1141:APEndometriumPr | 10935 | 104 | 90006 | 85449 | 84063 | 69447 | 77994 | **67263** |
| 1142:OVAEndometrium | 10935 | 1236 | 81989 | 74574 | 73086 | 81493 | 76062 | **69242** |
| 1143:APColonOmentum | 10935 | 290 | 84332 | 77198 | 77140 | **65250** | 76908 | 66816 |
| 1144:APProstateKidne | 10935 | 263 | 85277 | 79553 | 79553 | 69589 | 77857 | **68052** |
| 1145:APBreastColon | 10935 | 504 | 84416 | 77346 | 77305 | **65165** | 78144 | 68448 |
| 1146:OVAProstate | 10935 | 1236 | 81989 | 75318 | 73160 | 81542 | 76434 | **69266** |
| 1147:APOmentumKidney | 10935 | 269 | 84780 | 75114 | 74952 | 68796 | 77112 | **67770** |
| 1148:APBreastUterus | 10935 | 374 | 85500 | 73275 | 73155 | **66892** | 79575 | 68475 |
| 1149:APOvaryKidney | 10935 | 366 | 86062 | * | 76590 | 85470 | 77626 | **68835** |
| 1150:APBreastLung | 10935 | 376 | 86868 | 86032 | 86260 | **68195** | 80332 | 69441 |
| 1151:APEndometriumOm | 10935 | 110 | 89474 | 98868 | 100386 | 68178 | 80388 | **66550** |
| 1152:APProstateOvary | 10935 | 213 | 87118 | * | 75594 | 86473 | 78045 | **68026** |
| 1153:APColonOvary | 10935 | 387 | 85488 | * | 91572 | 85254 | 81198 | **68047** |
| 1154:APEndometriumLu | 10935 | 149 | 89490 | 74940 | 75180 | 70350 | 76710 | **67260** |
| 1233:eating | 6373 | 756 | 7843 | 6703 | 5381 | **-1110** | 7980 | 5457 |
| 1457:amazon-commerce | 10000 | 1200 | 15576 | 11376 | 11256 | **10680** | 12216 | 10920 |
| 1458:arcene | 10000 | 160 | 19181 | 9152 | 8746 | **-1267** | * | 8179 |
| 1484:lsvt | 310 | 100 | 200 | **152** | 872 | 212 | 464 | 180 |
| 1514:micro-mass | 1300 | 288 | 1166 | 50547 | 50912 | 1056 | * | **-708** |
| 1515:micro-mass | 1300 | 456 | 1260 | 8041 | 71493 | 1224 | * | **589** |
| Total # wins | | | 0 | 1 | 1 | 18 | 0 | **32** |

## Footnotes

[1] Unless specified all expectations are taken with respect to the joint distribution $p_W(x, z)$.

[2] One can set $Q_i \propto TC(Z|X_i)$. However, we choose not to do this since we do not have a derivation that leads the resulting objective into an equivalent, but efficiently computable objective.

[3] https://github.com/fdtomasi/regain