[Reviews · NeurIPS 2019]

Reviewer 1



The manuscript proposes a new objective function for learning Gaussian latent factor models. The objective function is based on information-theoretic characterization of modular latent factor models, where the model attains optimal value. The derivation of the objective function carefully avoids matrix inversion to improve computational complexity compared to traditional methods. The authors pointed out that the proposed model enjoys 'blessing of dimension' in that model performance improves when the dimension of observable variables increases while the dimension of latent variables remains constant. This is demonstrated by both simulation and an information-theoretic lower bound on the sample size. The authors also conducted extensive experiments to show the advantages of the proposed method in estimation accuracy and computation speed. Quality: The main results in the manuscript are supported by either theoretical analysis or experiments. I find the theoretical component of the manuscript solid in general. Although some mistakes exist in the technical details of the derivations/proofs (Items 5, 8, 9 below), they didn't affect the main results. The experiments are extensive in that a large number of datasets and competing methods are explored. However, using likelihood as evaluation metric doesn't seem appropriate in the real-world data analysis (Item 4 below). I think this is the main weakness of the manuscript. Clarity: The manuscript is well-written in general. It does a good job in explaining many results and subtle points (e.g., blessing of dimensionality). On the other hand, I think there is still room for improvement in the structure of the manuscript. The methodology seems fully explainable by Theorem 2.2. Therefore, Theorem 2.1 doesn't seem necessary in the main paper, and can be move to the supplement as a lemma to save space. Furthermore, a few important results could be moved from the supplement back to the main paper (e.g., Algorithm 1 and Table 2). Originality: The main results seem innovative to me in general. Although optimizing information-theoretic objective functions is not new, I find the new objective function adequately novel, especially in the treatment of the Q_i's in relation to TC(Z|X_i). Relevant lines of research are also summarized well in the related work section. Significance: The proposed methodology has many favorable features, including low computational complexity, good performance under (near) modular latent factor models, and blessing of dimensionality. I believe these will make the new method very attractive to the community. Moreover, the formulation of the objective function itself would also be of great theoretical interest. Overall, I think the manuscript would make a fairly significant contribution. Itemized comments: 1. The number of latent factors m is assumed to be constant throughout the paper. I wonder if that's necessary. The blessing of dimensionality still seems to hold if m increases slowly with p, and computational complexity can be still advantageous compared to GLASSO. 2. Line 125: For completeness, please state the final objective function (empirical version of (3)) as a function of X_i and the parameters. 3. Section 4.1: The simulation is conducted under a joint Gaussian model. Therefore, ICA should be identical with PCA, and can be removed from the comparisons. Indeed, the ICA curve is almost identical with the PCA curve in Figure 2. 4. In the covariance estimation experiments, negative log likelihood under Gaussian model is used as the performance metric for both stock market data and OpenML datasets. This seems unreasonable since the real data in the experiment may not be Gaussian. For example, there is extensive evidence that stock returns are not Gaussian. Gaussian likelihood also seems unfair as a performance metric, since it may favor methods derived under Gaussian assumptions, like the proposed method. For comparing the results under these real datasets, it might be better to focus on interpretability, or indirect metrics (e.g., portfolio performance for stock return data). 5. The equation below Line 412: the p(z) factor should be removed in the expression for p(x|z). 6. Line 429: It seems we don't need Gaussian assumption to obtain Cov(Z_j, Z_k | X_i) = 0. 7. Line 480: Why do we need to combine with law of total variance to obtain Cov(X_i, X_{l != i} | Z) = 0? 8. Lines 496 and 501: It seems the Z in the denominator should be p(z). 9. The equation below Line 502: I think the '+' sign after \nu_j should be a '-' sign. In the definition of B under Line 503, there should be a '-' sign before \sum_{j=1}^m, and the '-' sign after \nu_j should be a '+' sign. In Line 504, we should have \nu_{X_i|Z} = - B/(2A). Minor comments: 10. The manuscript could be more reader-friendly if the mathematical definitions for H(X), I(X;Y), TC(X), and TC(X|Z) were state (in the supplementary material if no space in the main article). References to these are necessary when following the proofs/derivations. 11. Line 208: black -> block 12. Line 242: 50 real-world datasets -> 51 real-world datasets (according to Line 260 and Table 2) 13. References [7, 25, 29]: gaussian -> Gaussian Update: Thanks to the authors' for the response. A couple minor comments: - Regarding the empirical version of the objective (3), it might be appropriate to put it in the supplementary materials. - Regarding the Gaussian evaluation metric, I think it would be helpful to include the comments as a note in the paper.

Reviewer 2



The paper shows surprising results that the simpler modular latent factor models can show better performance in covariance estimation than the more expressive latent factor models (e.g. PCA / factor analysis). I think this result is interesting and deserves more attention for future research. The experiments in the paper are extensive. They include a large number of data sets and two data sets for specific illustration. They also include an acceptable selection of methods for comparison. The experimental results are sufficiently convincing. The presentation is good in general. On the other hand, the paper is dense and therefore there may be difficulty in making a good selection of content. In the experiments, the synthetic data sets were generated from modular latent factor models or ones with similar structure. Therefore, it is not surprising that the proposed method could achieve the best results. As such, I think those experiments may not deserve so much space. The modular latent factor models under consideration are somewhat similar to latent tree models (such as the pouch latent tree models in [3], latent tree graphical models with continuous variables in [15], and the model in [*]). Hence, the experiments would be more interesting if those methods can be included for comparison. Those models may have different structures, or the latent variables may be discrete, but they should still be able for use in covariance estimation. Although the experiments with stock market data show some interesting results, it is not clear whether the methods used in comparison are those commonly used for such data. It would perhaps be better if the authors can cite related work in this area. The section on fMRI data may be too brief and it is not easy to appreciate the usefulness of the proposed method. The correlation explanation method seems to have been proposed by the authors of [40] and [41] before. The authors may explain the relation between the proposed method and the previous correlation explanation method(s) with more details (in contrast with the single sentence used in section 5 currently). Minor comments: 1) Why the two models “admit equivalent information-theoretic characterization”? The caption may explain this or what information-theoretic characterization refers to and also the meaning of TC. 2) The paper should cite [32] when it first discusses blessing of dimensionality (provided that the current paper is not the first to discuss this concept). 3) Lines 119-120 on page 3 may not be necessary. 4) The paper may state more clearly how many latent factors are used by the proposed method on the stock market data. [*] G. Galimberti, G. Soffritti, Model-based methods to identify multiple cluster structures in a data set, Comput. Statist. Data Anal. 52 (2007) 520–536.

Reviewer 3



Originality: The proposed method is original, and to the best of my knowledge the related work regarding latent models is adequately cited. Quality: The submission appears technically sound. I was able to check the main theoretical results, which to me are correct. The presented experimental results appear convinving, since they cover a wide range of problems. The authors discuss both strength and weaknesses of their approach, i.e., it assumes a jointly Gaussian distribution for correctness, and it suffers convergence issues with nearly-deterministic distributions. Clarity: I found the paper hard to read. First, the paper suffers from a very dense content, in particular extensive mathematical derivations and experiments, which the authors have decided to place in the supplementary materials. Also, a lot of statements are directly referring to content in the supplementary materials for justification, which brings the actual total length of the paper to 21 pages rather than 8. Still, the main 8-pages body remains understandable on its own. A second point, which I believe the auhtors can fix, is that the paper is quite confusing and even a bit misleading in several aspects, such as the time complexity of the approach, the exact goal of the approach (structure learning ? regularized density estimation ?), or some divergences between the theoretical presentation of the method and its implementation, that is, the factorization p(z|x) = \prod_j p(z_j|x), which are not discussed at all. See the detailled list of things that have to be fixed below: l.8-9: The proposed method has linear time complexity w.r.t. the number of observed variables -> This is misleading. The proposed criterion to be minimized can be computed in linear time, however the whole learning procedure to minimize this criterion is most likely not. l.18-19: is a holy grail in many scientific domains, including neuroscience, computational biology, and finance -> any reference ? l.22-29: For instance, graphical LASSO, [...] have better time complexity but perform very poorly in undersampled regimes. -> What does it mean to perform poorly ? For which task ? Density estimation ? What does it mean quantitatively ? At this point the text is very vague. Please state from the start which tasks those methods or your proposed method intend to solve, and which criterion permits to compare them objectively. l.30-31: a novel latent factor modeling approach for uncovering the structure of a multivariate Gaussian random variable -> This is very confusing. Your approach not only learns the structure of a multivariate distribution, but also does density estimation. There exists a whole litterature in graphical model structure learning which learns the structure only (i.e., just the graph), see [Murphy 2012, Ch.26]. Please clarify this, e.g., "a novel latent factor modeling approach for estimating multivariate Gaussian distributions with modular structures". l.33-34: when each observed variable has a single latent variable as its only parent -> You are suddently talking about parents here, but at this point you didn't state yet that you consider directed graphical structures. There exists undirected structures as well, which can also be latent models. Please clarify. l.36: is block-diagonal with each block being a diagonal plus rank-one matrix -> Why would each block be rank-one ? It is not obvious to me, I first figured this was an assumption you were making. Is that statement important for the paper ? l.36-38: This modular inductive prior is appropriate for many real-world datasets, such as stock market, magnetic resonance imaging, and gene expression data -> Why would that be ? Any intuition behind that ? Evidence ? References ? Figure 1: (Thm. 2.2) / (for Gaussians) -> This is very misleading, since Theorem 2.2 assumes a Gaussian distribution. Or reformulate Theorem 2.2 as "a modular latent factor implies TC(Z|Xi), and the reverse is true for Gaussian distributions". l.40: gets easier -> in terms of what ? Time ? Log-likelihood ? Structural error ? l.44: unconstrained graphical models -> unconstrained latent factor models l.68: constrains -> constraints Theorem 2.1: This is trivial, and is more a definition than a theorem. Random variables X and Z define a latent factor model iff. p(x,z) = \prod_j p(z_j) \prod_i p(x_i|z) <=> TC(Z) + TC(X|Z) = 0. Definition 2.1: i=1,2,...,p is redundant with \prod_{i=1}^p l.89: sometimes called clustering of observed variables -> reference ? l.93: from [7] -> yhis is bad style. Please name the authors or the approach. l.98: jointly -> multivariate ? Please be consistent in the text. Theorem 2.2: Fig. 1b equivalence -> this is a very bad name for a Theorem. For the sake of readability, either do not give a name or give a proper name, e.g, Modular Latent Gaussian Characterization. l.108: jointly Gaussian -> This is not a general parameterization, but rather a constrained Gaussian distribution where TC(Z|X) = 0. How does that interplay with TC(Z) = 0, TC(X|Z) = 0 and TC(Z|Xi) = 0 ? Is this important for Linear CorEx to work ? Please mention at least that this additionnal constraint is compatible with a modular latent model. This was very confusing to me. Equation (1): minimize_{z=Wx+e} -> minimize_{W} l.145: by increasing m until the gain in modeling performance is insignificant -> How do you measure a gain ? Log-likelihood ? l.148: The computational complexity -> The stepwise computational complexity Figure 3: in both cases s=5 -> Either say what s is here (signal-to-noise ratio), or don't mention it. Knowing what s means requires digging the text. l.208: black -> block l.208: each black being a diagonal plus rank-one matrix -> Why is it rank-one ? Why is that important ? l.246-247: we chose the number of factors [...] -> using which criteria ? Log-likelihood ? l.252-253: We omitted empirical covariance estimation since all cases have n < p -> Why ? I would still be curious to have the numbers to compare. l.271: session 014 -> Why ? this sounds a lot like cherry-picking. l.272-273: We do spatial smoothing by applying a Gaussian filter with fwhm=8mm -> Why ? Your method doesn't work if you don't do that ? Significance: The presented results are important, as they provide 1) an efficient way of learning modular latent Gaussian models, and 2) empirical evidence that this modular prior results in a better density estimation than other state-of-the-art approaches on a wide range of problems. Some general comments which the authors may want to address: - Suppose the X distributions contains variables Xi which are independent of the rest, i.e., X_i \ind X\{X_i}. Then such variables may not be linked to any latent variable, and still your minimized criterion can reach 0. The learned structure is a modular latent factor model. How do you deal with that situation in variable clustering ? Your procedure picks the latent variable with highest correlation, yet all correlations with that particular variable are 0. An easy answer is to assign each such variable to its own cluster. You may want to discuss that in the text. - You can simplify the proof for Theorem 2.2, and at the same time make it more general than only Gaussian distributions. Indeed, Gaussian distributions, among others, support the Decomposition, the Composition and the Singleton-Transitivity properties (See, e.g., Sadeghi 2017, Faithfulness of Probability Distributions and Graphs, p.7). Then you have: Z_s \indep Z\Z_s for all Z_s \subseteq Z (<=> TC(Z) = 0) Z_s \indep Z\Z_s | X_i for all Z_s \subseteq Z and X_i \in X (<=> TC(Z|X_i) = 0) Then due to the Singleton-Transitivity: Z_s \indep X_i or Z\Z_s \indep X_i Either both sides are true, in which case X_i \indep Z due to the Composition, and Z_s \indep X_i for all Z_s \subseteq Z due to the Decomposition. Or one side is false, for all Z_s \subseteq Z, in which case the intersection of all such Z_s or Z\Z_s sets leaves out a single Z_j, the unique parent of X_i

[Author Response · NeurIPS 2019]

We would like thank the reviewers for the detailed comments and suggestions, which we believe will improve the manuscript.

**Reviewer #1.** **1.** The sample complexity lower bound proved in Thm. C.1 shows that for large values of $p$, $n$ grows like $\log m$ (please see the inequality below line 543). Consequently, if $m$ is a growing function of $p$, no method will exhibit blessing of dimensionality. **2.** We would wish to write down the empirical version of the final objective (3) as a function of the inputs and parameters nicely in a single line. However, it requires introducing notation for samples and adding many repetitive equations for estimating $R_{j,i}, B_{j,i}, r_i$, and $\nu_{X_i|Z}$. **4.** We agree that Gaussian log-likelihood is not the best evaluation metric when the data is non-Gaussian. However, we would like to note that in our experiments most of the baselines, such as glasso, latent variable glasso, [sparse] PCA, are derived under Gaussian assumption. **7.** The "law of total variance", should be "law of total *covariance*". We apologize for the confusion, and will fix this in the camera-ready version. **10.** We will add definitions of quantities such as $H(X), I(X:Y), TC(X), TC(X|Z)$ in the notation paragraph of section 2. **5,6,8,9,11-13.** These flaws and issues will be fixed in the camera-ready version.

**Reviewer #2.** **1.** With the experiments on synthetic data generated from a modular latent factor model, we primarily wanted to demonstrate that the proposed method exhibits blessing of dimensionality. Another goal was to show that, perhaps unsurprisingly, linear CorEx outperforms other methods in covariance estimation. We decided to have this latter result in the main text, since it also demonstrated how the performance gap changes with the number of samples and sparsity level of the ground truth covariance matrix. **2.** We agree that the set of baselines we considered in the stock market experiments probably does not contain the best methods for that specific task. The goal of that experiment was not to show state-of-the-art results on stock market data, but to compare the proposed method with other general covariance estimation methods on a useful real-world dataset. Nevertheless, we look forward to convey this more clearly and cite relevant literature. **3.** In the experiment on fMRI data, the goal is the demonstrate that linear CorEx scales for voxel-level analysis of brain data, while producing meaningful results. We believe the potential of our method for analysing brain data is yet to be explored, with more careful experiments that will contain multiple patients or multiple sessions. **4.** Regarding to the comment on correlation explanation methods proposed by authors of [40] and [41], we will elaborate on the relation of our method with those methods.

**Reviewer #3.** We are happy to fix misleading aspects of the manuscript and improve its clarity. **1.** We will make it explicit that is the stepwise computational complexity of the method that is linear w.r.t the number of variables. **2.** We agree that the exact goal of the approach – finding a latent factor model that is close to being modular – is not stated clearly. We apologize for this and will fix this. **3.** As the reviewer correctly points out, when parameterizing $p_W(z|x)$ we make an implicit assumption that $Z$s are conditionally independent given $X$. We make this assumption since it simplifies the further derivations and is respected by modular latent factor models. This note will be added in the camera-ready version.

Below are our clarifications to your questions. Please note that these clarifications and your suggestions will be added in the camera-ready version.

36, 208 Modular latent factor models have block diagonal covariance matrices. Additionally, each block is a diagonal plus rank-one matrix. To see this, consider a block where $Z$ is the parent and $X_1, X_2, \ldots, X_p$ are the children with respectively $\rho_1, \rho_2, \ldots, \rho_p$ Pearson correlation coefficients with $Z$. Then, $\text{Cov}[X_i, X_{j\neq i}] = \sigma_i\sigma_j\rho_i\rho_j$, where $\sigma_i^2 = \text{Var}[X_i]$. We decided to mention this property of covariance matrices of modular latent factors to demonstrate which types of covariance matrices are preferred by the objective of our method. We though this would be helpful for comparing our method with [latent variable] glasso.

36-38 The intuition is that in those types of data variables can be divided into clusters, where each cluster is governed by a few latent factors and latent factors of different clusters are close to be independent.

40 Learning becomes easier in terms of structural error.

145 We measure the gain by log-likelihood.

246-247 We choose the number of factors according to their log-likelihood score on the validation data.

252-253 Empirical covariance matrices in those cases are not invertible and computing negative log-likelihood becomes impossible.

271 Session 014 is the first publicly available session of that dataset.

272-273 Spatial smoothing intensifies correlations between nearby voxels, helping our model to pick-up the spatial information faster. Without spatial smoothing the training is unstable and we suspect that more samples are needed to train the model.

[Meta-Review · NeurIPS 2019]

This manuscript proposes an estimator for graphical models which encourages modularity. The strengths of the manuscript include the conceptual simplicity of the proposal and the clear analysis. Reviewers also commented on the overall clarity of the presentation and the extensive experiments. I encourage the authors to read the reviews carefully and make changes as appropriate for the final version.